# Q-Adapt: Adapting LMM for Visual Quality Perceiver with Progressive Instruction Tuning

## Abstract

The rapid advancement of Large Multi-modal Foundation Models (LMM) has paved the way for the possible Explainable Image Quality Assessment (EIQA) with instruction tuning from two perspectives: overall quality explanation, and attribute-wise perception answering. However, existing works usually overlooked the conflicts between these two types of perception explanations during joint instruction tuning, leading to insufficient perception understanding. To mitigate this, we propose a new paradigm for perception-oriented instruction tuning, *i.e.*, Q-Adapt, which aims to eliminate the conflicts and achieve the synergy between these two EIQA tasks when adapting LMM, resulting in enhanced multi-faceted explanations of IQA. Particularly, we propose a progressive instruction tuning strategy by dividing the adaption process of LMM for EIQA into two stages, where the first stage empowers the LMM with universal perception knowledge tailored for two tasks using an efficient transfer learning strategy, *i.e.*, LoRA, and the second stage introduces the instruction-adaptive visual prompt tuning to dynamically adapt visual features for the different instructions from two tasks. In this way, our proposed Q-Adapt can achieve a lightweight visual quality perceiver, demonstrating comparable performance and, in some instances, superior results across perceptual-related benchmarks and commonly-used IQA databases.

## 1 Introduction

Image Quality Assessment (IQA) aims to evaluate whether the image fidelity satisfies the human visual experience (Moller et al., 2009; Reiter et al., 2014), which has been used to various image processing techniques such as image compression (Yang & Mandt, 2024; Wu et al., 2021), restoration (Liang et al., 2021; Xia et al., 2023). However, despite that most IQA metrics, *e.g.*, DEIQT (Qin et al., 2023), LIPIPS (Zhang et al., 2018) can provide an accurate quality score, they cannot explain the reasons in terms of distortions and contents behind the corresponding score. With the advancement of Large Multi-modal Foundation Models (LMM), Explainable Image Quality Assessment (EIQA) has become feasible due to the multi-modal reasoning and interaction capabilities of LMMs. A series of preliminary attempts have been made to excavate the low-level perception capability for images using LMMs (Wu et al., 2023a; Zhu et al., 2024; Wu et al., 2023b).

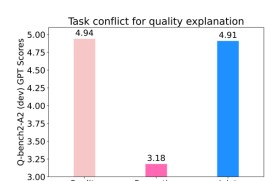

Figure 1: The effect of different task instruction tuning for quality explanation task.

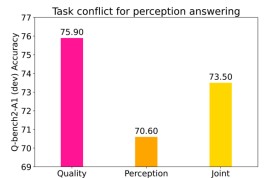

Figure 2: The effect of different task instruction tuning for perception answering task.

Existing works on LMM-based IQA can be roughly divided into two types. The first type aims to adapt the pre-trained LMMs to downstream IQA tasks by designing prompt templates, *i.e.*, prompt engineering, while freezing the parameters of LMMs. For instance, simply quality-aware prompt design can enable the GPT-4V (OpenAI, 2023; Wu et al., 2023a; Zhang et al., 2024) with great low-level visual perception capability. Despite the efficient adaptation, the frozen parameters limit the adequate low-level perception knowledge excavation required by downstream IQA tasks. The second type of works (Wu et al., 2023b; 2024a; You et al., 2023) relies on instruction tuning, which aims to empower the pre-trained LMMs with overall quality explanation capability (*i.e.*, the left part of Fig. 3) and attribute-wise perception answering (*i.e.*, the right part of Fig. 3) capability by tuning the LMMs,

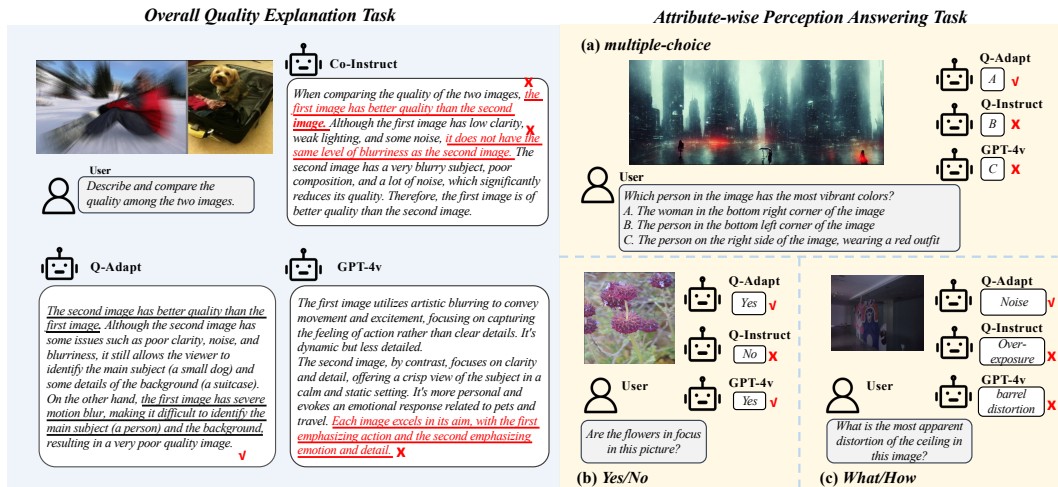

Figure 3: The comprison between existed LMMs and our proposed Q-Adapt on two EIQA tasks (*i.e.*, the overall quality explanation task, and the attribute-wise perception answering task). Our proposed Q-Adapt can generate more accurate response, benefiting from the reduction of task conflicts and the enhanced synergy between the two tasks, achieved through progressive instruction tuning.

preliminarily bridging the path to explainable IQA from two explanation perspectives. From Fig. 1 and Fig. 2, we observe that focusing exclusively on the explanation task improves performance compared to joint tuning of both tasks. Additionally, as illustrated in Fig. 3, Co-Instruct and GPT-4V exhibit instances of visual hallucinations in the question answering task. These observations highlight two fundamental challenges in LMM-based explainable image quality assessment: (i) The conflicts between these two EIQA tasks are overlooked during instruction tuning, caused from the bias towards attribute-wise perception knowledge and the degradation of universal perception knowledge. (ii) The insufficient cross-modal interaction restricts the adaptability to the synergy between these two EIQA tasks. As Fig. 3 illustrates, insufficient reasoning capability and inflexible task instruction adaptation lead to misleading and spurious responses.

To address the above issues, we propose Q-Adapt, a new paradigm for perception-oriented instruction tuning. Q-Adapt aims to eliminate task conflicts and achieve synergy between the two EIQA tasks, thereby enhancing the multifaceted explanations of IQA when adapting LMM as visual quality perceiver. Specifically, we propose a progressive instruction tuning by dividing the adaptation process of LMM for EIQA into two stages, continously enhancing perception knowledge for both tasks. The first stage involves the acquisition of universal perception knowledge in a parameter-efficient manner (*i.e.*, LoRA (Hu et al., 2021)), establishing a powerful foundation that supports the different instruction requirements of both EIQA tasks. Building on the universal perception knowledge acquired in the first stage, we can more easily achieve adaptability for instructions across different tasks. However, the limited multimodal interactions (Dong et al., 2024) within the layers of the LMM's language decoder are insufficient for adaptively capturing the visual knowledge specified by the instructions across both tasks. To overcome this dilemma, we introduce instruction-adaptive visual prompt tuning, which dynamically adapts visual features to the different instructions, thereby enhancing the synergy between the two EIQA tasks. In particular, to develop a visual prompt with powerful instruction adaptive capabilities, we employ bi-directional multimodal interactions to obtain an instruction-adaptive visual prompt, which consists of a vision-text (V-T) generator to fuse perception-related visual knowledge required by instructions into textual feature, and a text-vision (T-V) prompter that projects the textual feature back into the visual space. The obtained instruction-adaptive visual prompt can guide the original visual feature through gated residual addition to highlight the crucial information specified by different instructions. Unlike uni-directional multimodal interactions (e.g., Q-Former (Dai et al., 2024)), which capture condensed semantic information (Yao et al., 2024) but lose fine-grained visual details, our bi-directional multimodal interaction module effectively acquires task-adaptive visual knowledge and refines the original visual feature without losing visual details. In summary, the contributions of this paper are summarized as follows:

- We point out that simultaneously tuning LMMs with two types of Explainable Image quality Assessment (EIQA) tasks (*i.e.*, overall quality explanation and attribute-wise perception answering), can lead to potential task conflicts and insufficient perception understanding.

- To alleviate the above task conflicts, we introduce a new paradigm for perception-oriented instruction tuning, namely Q-Adapt. Q-Adapt employs a progressive instruction tuning which consists of two stages for adaption process for LMM: the universal perception knowledge learning stage and the instruction-adaptive visual prompting stage. This approach achieves synergy between the two EIQA tasks and enhances the multifaceted explanations of IQA.

- Experimental results on perceptual-related benchmarks and commonly-used IQA databases demonstrate that Q-Adapt achieves comparable and in some cases superior, performance, even when utilizing a lightweight LMM model (*i.e.*, Bunny-3B (He et al., 2024)).

## 2 RELATED WORK

**Large Multimodality Foundation Model** The large language models (LLM) have shown the powerful ability to act as a universal interface for a general-purpose assistant (Zhang et al., 2023b). Following the step of LLM, LMMs are extended to conduct visual language tasks, which have achieved remarkable progress in multiple visual recognition and reasoning tasks (Chen et al., 2023a; Peng et al., 2023; Ren et al., 2023; Liu et al., 2023). The cutting-edge works (Liu et al., 2023; Li et al., 2022; Dai et al., 2024) of LMM mainly bridge the visual encoder and LLM with a cross-modality connector to achieve the multimodal understanding ability. The milestone achievement, LLava (Liu et al., 2023) introduces visual instruction tuning to advance towards a general-purpose assistant. And the following works in LMM can be divided into two categories: i) enhance visual perception, ii) enhance the interaction between visual and text representation. For the first category, current works primarily optimize the visual representation by scaling the visual extractor or combining multiple visual experts. From the perspective of the parameter scale of visual encoder, InternVL (Chen et al., 2023b) scales up the visual encoder to match the parameter scale of LLM and proposes a progressive alignment strategy to harmonize the multimodal representations, which achieves outstanding ability in many vision-language tasks. Due to the limitation of CLIP visual encoder, Tong *et. al* (Tong et al., 2024) interleaves the image feature from CLIP visual encoder and DINO (Caron et al., 2021; Darcet et al., 2023) to enhance the visual grounding capabilities. Sphinx (Lin et al., 2023) mixes image features from various visual encoders to achieve a versatile visual understanding ability. As for the second category, existing methods primarily focus on aligning visual and textual features before feeding into LLM, or conducting visual-text collaboration/interaction within the deeper layers of the LLM. To align visual features with task-specific instructions, InstructBLIP (Dai et al., 2024) excavates the instruction-aware multimodal feature through Q-Former before integration into the LLM. To implement multimodal collaboration, mPLUG-Owl2 (Ye et al., 2023) processes visual and text features through different modules in each layer of LLM. With the same inspiration, CogVLM (Wang et al., 2023b) inserts the visual expert in each layer of LLM for deep alignment between two modalities. Inspired by the above two improvements, we aim to enhance the task-instruction adaptability of visual representation for multi-modal shallow alignment, thereby enabling the adaptive selection of the required granularity of perceptual knowledge to facilitate the reasoning process in LLMs.

**Large Multimodal Foudation Model for IQA** LMM for Image Quality Assessment (IQA) can be divided into tree main streams. The first is to apply LMM to align the quality feature into text space. LIQE (Zhang et al., 2023c) fine-tuned the CLIP (Radford et al., 2021) model with fidelity loss to perceive the semantic-level scene, low-level distortion, and quality-level score. Inspired by prompt learning for CLIP (Zhou et al., 2022), CLIPIQA (Wang et al., 2023a) assesses quality scores by constructing prompt pairs with antonyms to evaluate the model's preference probability for score tokens. Through text generation, Q-Align (Wu et al., 2023c) enables LMM to evaluate quality scores that align with human opinions. The second is using the prompt engineering technique to activate the quality perception ability of LMM. Zhu *et. al* (Zhu et al., 2024) employ two alternative forced choice (2AFC) prompting for multiple LMMs to explore their quality assessment ability. To study more prompt strategy on LMM for quality assessment, Wu *et. al* (Wu et al., 2024b) explores the chain-of-thought, in-context prompt to conduct the pair-wise image quality comparison. The third is to activate the instruction-following ability of LMM for explainable image quality assessment (EIQA). This line of research begins with the development of fine-grained low-level perceptual-related benchmark (Huang et al., 2024; Wu et al., 2023a; Zhang et al., 2024), to evaluate the performance of both open-source (Zhu et al., 2023; Zhang et al., 2023a; Ye et al., 2023; Liu et al., 2023) and proprietary large multimodal models (OpenAI, 2023; Google, 2023). Subsequently, it involves the

creation of the instruction datasets (Wu et al., 2023b; 2024a) that consists of the overall quality explanation task and attribute-wise perception answering task. These efforts aim to enhance the instruction-following ability of advanced multimodal large models for low-level vision. These approaches bridge the existing gap in IQA models regarding the capability for textual reasoning and interaction in an explainable manner. In contrast to these approaches, our method facilitates the adaptation of LMMs to visual quality perception through efficient training. By mitigating the conflicts between the two EIQA tasks, we aim to achieve a more comprehensive understanding of visual quality perception.

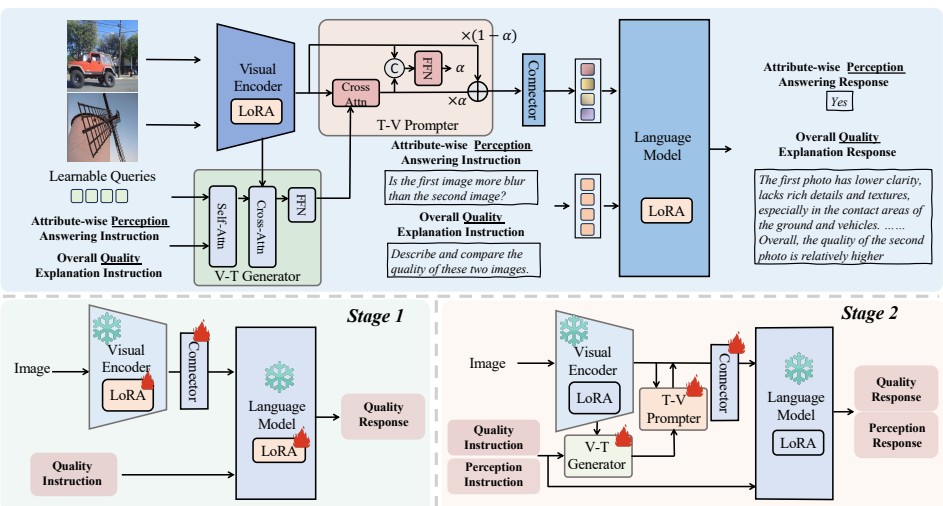

Figure 4: The overview for our proposed Q-Adapt, which employs progressive instruction tuning to achieve the synergy between two EIQA tasks. Concretely, the progressive instruction tuning strategy comprises two stages: the universal perception knowledge requiring stage (*i.e.*, the first stage) tailored for building a powerful base for two tasks, and the instruction-adaptive visual prompting stage for dynamically adapting visual features for task instruction. Additionally, the second stage incorporates the V-T Generator and T-V Prompter to achieve the bi-directional multimodal interactions.

## 3 METHOD

### 3.1 PRELIMINARIES

The primary objective of the Large Multi-modality Foundation Model (LMM) is to perceive visual signals and engage in reasoning through interactions with textual instructions, thereby addressing a variety of visual-language tasks. The structure of the current LMM can be primarily summarized into three parts: the visual encoder, large language model (LLM), and multi-modal connector for bridging the visual and textual modality. As for Explainable Image Quality Assessment (EIQA) task, given an image $v$ and perceptual-related instruction $I$, we extract the image feature $F_v \in \mathbb{R}^{n \times d_v}$ through the visual encoder, where $n$ is the number of visual tokens, and $d_v$ is the channel dimension. These features are subsequently processed through a connector $f_{vt}$, which maps them into the textual space, resulting in $F_{vt} \in \mathbb{R}^{n \times d_t}$, where $d_t$ represents the channel dimension, aligning with that of the text tokens.. The transformed features, along with the instruction embedding $F_I \in \mathbb{R}^{m \times d_t}$, where $m$ denotes the number of the text tokens, are then fed into the Large Language Model (LLM). Optimization is performed using a language modeling loss based on next-token prediction (Liu et al., 2023; Touvron et al., 2023), which models the likelihood of the generated response conditioned on the provided images and instructions:

$$L(r, v, I) = - \sum_{l=1}^{L_1} \log \left( P(r_l | v, I, r_{<l}) \right) \tag{1}$$

Where $r_l$ represents the generated response token, conditioned on the input image $v$, instruction $I$, and previously generated response tokens $r_{<l}$.

## 3.2 Task Conflicts for Explainable Image Quality Assessment

The Explainable Image Quality Assessment (EIQA) contain two tasks (Wu et al., 2023a): overall quality explanation (Wu et al., 2023b; You et al., 2023), and attribute-wise perception answering (Wu et al., 2023b; 2024a). As shown in Fig. 3, The first task requires a long-text response detailing an overall quality explanation that integrates multiple low-level attributes and concludes with a final quality score. The second task includes three types of perceptual-related visual question answering: multiple-choice, yes/no, and what/how questions, requiring brief answers for specific attributes/dimensions. From Fig. 2, we observe that tuning solely on the overall quality explanation task results in increased performance in the attribute-wise perception answering task, when compared to joint tuning on two tasks. It indicates that (i) an inherent conflict exists between the two tasks, since attribute-wise knowledge derived from training on the perception answering task tends to narrow the focus of the LMM towards localized/specific dimensions, lacking universal reasoning ability; (ii) the universal perception knowledge acquired through training on the quality explanation task explicitly assists in enhancing the reasoning capabilities for visual quality perception, which can build a powerful foundation for the two tasks.

## 3.3 Progressive Instruction Tuning

### 3.3.1 Universal Perception Knowledge Learning Stage

To address the conflicts between the two EIQA tasks, we introduce the progressive instruction tuning strategy to enhance perception knowledge for the two EIQA tasks. It consists of two stages for perceptual-related instruction tuning on two tasks. Based on the above observation, we are inspired to utilize the universal perception knowledge acquired from the overall quality explanation task to facilitate subsequent task adaption for different instructions. Therefore, the first stage involves the instruction tuning on the quality explanation tasks for universal perception knowledge acquisition. To effectively learn the universal perception knowledge, this stage involves fine-tuning with a multimodal connector and utilizing the parameter-efficient LoRA (Hu et al., 2021) technique on both the LLM and visual encoder. Specifically, the loss function of stage1 can be formulated as:

$$L_{\text{stage1}}(a_q, v, I_q) = -\sum_{l=1}^{L_1} \log\left(P_{\Phi_0 + \Delta\Phi(\theta)}(a_{q,l}|v, I_q, a_{q,<l})\right) \tag{2}$$

where $\Phi_0$ and $\Delta\Phi(\theta)$ are referred to the parameters of frozen LMM and learnable LoRA parameters, respectively. And the subscript $q$ denotes the overall quality explanation task. $a_{q,l}$ represents the $l$-th token of the answer, and $I_q$ denotes the instruction of the overall quality explanation task. The $a_{q,<l}$ represents the generated answer token.

### 3.3.2 Instruction-guided Visual Prompt Tuning Stage

In the second stage, to effectively enhance the perceptual knowledge for two EIQA tasks, two critical conditions must be fulfilled: (i) It is essential to adaptively select the required perception knowledge based on task instructions, which can alleviate the conflicts between the above two tasks. (ii) It is vital to ensure that the universal perception knowledge is not compromised by the attribute-wise knowledge from the attribute-wise perception answering task, thus enhancing the optimization of both tasks. Therefore, this stage requires fixing the parameters of the LLM and visual encoder, with the connector trainable, to prevent interference from biases towards specific perceptual knowledge for the single/localized dimension.

Also, the self-attention mechanism in the LLM decoder treats visual and textual tokens equivalently across all layers (Dong et al., 2024), which limits its flexibility in extracting task-specific knowledge from visual features due to the insufficient cross-modal interactions. Therefore, we propose the instruction-adaptive visual prompt tuning to excavate the essential knowledge required for the instruction for specific tasks. Concretely, we utilize the bidirectional interaction between instruction and visual features, which results in a prompt module comprising two specialized components: the V-T Generator, designed for vision-to-text interaction, and the T-V Prompter, tailored for text-to-vision interaction.

**V-T Generator** Due to the powerful vision-text interaction ability of the cross-attention-assisted transformer (*e.g.*, Q-Former) (Dai et al., 2024), we leverage the Q-Former to enhance instruction

representation with visual feature, enabling it to focus on informative visual knowledge for task instruction. Specifically, we input both the instruction representation $F_t$ and a fixed number of learnable queries $Q$ into the Q-Former. This process yields an instruction representation $F_t$ that is enriched with visual features $F_v$, effectively bridging visual and textual representations and injecting the visual knowledge related to the instructions. The formulation of Q-Former is listed as follows:

$$F_{vt} = \mathcal{G}(Q, F_t, f(F_v)) \tag{3}$$

Where, $Q \in \mathbb{R}^{m,d}$ denotes the learnable queries, $f(F_v)$ represents the projection for visual feature $F_v \in \mathbb{R}^{n,d_v}$ to match the dimension $d$. And the final obtained visual-guided instruction feature is $F_{vt} \in \mathbb{R}^{m,d}$. The V-T Generator, based on Q-Former, extracts instruction-adaptive visual features and maps them into the textual space, aggregating highly compressed perceptual information (Yao et al., 2024) via a limited number of learnable queries, which results in a loss of fine-grained visual details. We then employ T-V Prompter to refine the original visual features, enabling the dynamic capture of task-related perceptual knowledge.

**T-V Prompter**  To enhance the knowledge adaptation of the original visual features, we introduce a second stage of text-vision interaction. As depicted in Fig. 10, this stage employs a gated fusion process to generate an instruction-adaptive visual prompt. Specifically, we utilize cross-attention to integrate the information from highly-condensed multimodal feature $F_{vt}$ into the original visual feature $F_v$, facilitating the dynamic modulation of the original visual feature. Subsequently, a sigmoid-gated fusion mechanism is applied to merge the intermediate feature $\tilde{F}_{tv} \in \mathbb{R}^{n,d_v}$ with the original visual feature $F_v \in \mathbb{R}^{n,d_v}$.

$$\tilde{F}_{tv} = \mathrm{CA}(F_v, f(F_{vt}), f(F_{vt})) \tag{4}$$

$$F_{tv} = (1 - \sigma([f(\tilde{F}_{tv}), f(F_v)]))\tilde{F}_{tv} + \sigma([f(\tilde{F}_{tv}), f(F_v)])F_v \tag{5}$$

Where $f(\cdot)$ is utilized to map the channel dimension $d$ of $F_{vt}$ to $d_v$. CA denotes the cross attention mechanism between $F_v$ and $f(F_{vt})$. And $\sigma(\cdot)$ computes the weights for gated fusion. Through the above operations, we can modulate the original visual features through the gated residual addition, effectively integrating the instruction-adaptive visual prompt to refine the original visual feature. Therefore, the optimization loss for the second stage can be updated as follows:

$$L_{\text{stage2}}(a, v, I) = -\sum_{l=1}^{L_1} \log \left( P_{\Phi_2}(a_{q,l}|v, I_{q,<l}, a_{q,<l}) \right) - \sum_{l=1}^{L_2} \log \left( P_{\Phi_2}(a_{a,l}|v, I_{a,<l}, a_{a,<l}) \right) \tag{6}$$

where, $\Phi_2 = \Phi_1 + \Theta_p$, $\Theta_p$ is denoted as the parameters of our learnable prompt modules.

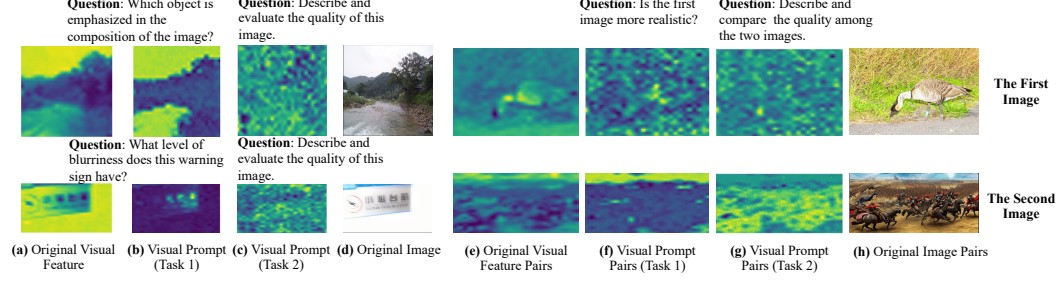

Figure 5: The visualizations of the original visual feature and instruction-adaptive visual prompt. (a)-(d) illustrate results from Q-bench, while (e)-(h) show results from Q-bench2. And "Task1" refers to the attribute-wise perception answering task, "Task2" denotes the overall quality explanation task.

## 4  EXPERIMENT

### 4.1  DATASETS AND IMPLEMENTATION DETAILS

**Training Datasets**  We conduct the perceptual-oriented visual instruction tuning on two datasets: Q-Instruct (Wu et al., 2023b) and Co-Instruct (Wu et al., 2024a). Q-Instruct has a total of 200k instruction-response pairs. Besides, Co-Instruct extends Q-Instruct from single image to multiple images, which includes a rich set of 580k instruction-response pairs. The model trained on Q-Instruct and Co-Instruct is named Q-Adapt$^Q$, Q-Adapt$^{Co}$, respectively.

**Evaluation Benchmarks** We evaluate our proposed Q-Adapt on the challenging perceptual-related benchmark Q-bench-A1 (Wu et al., 2023a) and Q-bench2-A1 (Zhang et al., 2024) for the attribute-wise perception answering task, and Q-bench2-A2 (Zhang et al., 2024) for the overall quality explanation task. We also tested the performance of our Q-Adapt on several commonly-used IQA datasets for image quality assessment (Hosu et al., 2020; Fang et al., 2020; Ying et al., 2020; Ghadiyaram & Bovik, 2015; Li et al., 2023a; Zhang et al., 2023d; Lin et al., 2019).

**Implementation Details** To construct the V-T Generator module, the Q-Former module in Instruct-BLIP (Dai et al., 2024) is applied as our V-T Generator. The number of queries in V-T Generator is 32, which follows previous work. And the cross-attention in T-V Prompter only has a single head.

Given that LMM is often constrained by their substantial computational costs and model parameters, we have adopted Bunny-3B (He et al., 2024), one of the lightweight multimodal model families for instruction tuning. The training of Q-Adapt requires two 32G V100 GPUs for training, and one 32G V100 GPU for testing. More details can be found in the **Appendix** A.

Table 1: Comparison of Different Methods for attribute-wise perception answering task.

| Method | Q-bench-A1 (%) | | | Q-bench2-A1 (%) | | |
|---|---|---|---|---|---|---|
| | dev | test | Average | dev | test | Average |
| Bunny-3B (He et al., 2024)*(Baseline)* | 65.08 | 64.68 | 64.88 | 48.20 | 50.85 | 49.53 |
| LLaVA-v1.5-13B (Liu et al., 2023) | 62.14 | 61.40 | 61.77 | 49.85 | 52.05 | 50.95 |
| mPLUG-Owl2 (Ye et al., 2023) | 61.61 | 62.68 | 62.15 | 49.85 | 48.94 | 49.40 |
| Emu2-Chat (Sun et al., 2023) | 65.28 | 64.32 | 64.80 | 50.05 | 47.08 | 48.57 |
| Qwen-VL-Max (Bai et al., 2023) | 73.63 | 73.90 | 73.77 | 67.27 | 66.99 | 67.13 |
| Gemini-Pro (Google, 2023) | 68.16 | 69.46 | 68.81 | 57.64 | 60.46 | 59.02 |
| GPT-4V (OpenAI, 2023) | 74.51 | 74.10 | 74.31 | 76.52 | 78.07 | 77.30 |
| Co-Instruct-8B (Wu et al., 2024a) | 76.99 | 77.12 | 77.05 | 78.40 | 80.18 | 79.29 |
| Q-Adapt-3B$^{Co}$ | 76.05 | 76.12 | 76.08 | 77.20 | 78.38 | 77.79 |
| Q-Instruct-8B (Wu et al., 2023b) | 70.23 | 73.38 | 71.81 | 50.54 | 53.15 | 51.85 |
| Q-Adapt-3B$^{Q}$ | 77.19 | 77.06 | 77.12 | 55.40 | 55.96 | 55.68 |

## 4.2 COMPARISON RESULTS

To verify the effectiveness of our proposed method, we evaluate our proposed Q-Adapt against two types of Large Multi-modal Foundation Models (LMMs): a frozen-based LMM and an instruction-tuning-based LMM. Some of frozen-based models (*e.g.*, GPT-4V (OpenAI, 2023), Gemini-pro (Google, 2023) and Qwen-max (Google, 2023)) are proprietary and closed-source. The performance of most of these frozen-based LMMs is generally inferior as they have not been exposed to image-quality-related textual data during previous training. Notably, within these comparative methods, **our Q-Adapt employs a parameter-efficient tuning strategy, and the total parameter size is only 3B.**

**Attribute-wise Perception Answering Task.** The results of performance comparison on the perception answering task are shown in Table 1. For Q-bench-A1, Q-Adapt$^{Q}$ surpasses the second-best method, Q-Instruct-8B, by a margin of 5.31% on average accuracy. And our Q-Adapt$^{Co}$, with a parameter size of 3B and LoRA training, achieves performance close to Co-instruct-8B on Q-bench2-A1.

**Overall Quality Explanation Task.** For Q-bench2-A2, the comparison results are represented in Table 2. Our Q-Adapt$^{Co}$ achieves a performance gain of 0.09 over the second-best method GPT-4V on the GPT score. It is attributed to our ability to achieve synergy between the two EIQA tasks, thereby improving perception precision. More examples can be found in **Appendix** A.3.

**Image Quality Assessment.** We also evaluate the performance of Q-Adapt$^{Q}$ on multiple IQA databases and compare it with existing LMMs and IQA models. For IQA models, LIQE (Zhang et al., 2023c) and LoDa (Xu et al., 2024) utilize networks to regress predicted scores against quality annotations. We transform the Q-Instruct dataset from image-text pairs to image-score pairs to facilitate regression for both LoDa and LIQE. From Table 3, Q-Adapt$^{Q}$ can achieve the best performance compared to other methods on the average performance of SROCC/PLCC. It is noteworthy that our Q-Adapt significantly outperforms existing LMMs and quality assessment models on the

Table 2: Performance comparison on overall quality explanation task. We employ the 5-round GPT score as defined in (Zhang et al., 2024) for our evaluation metric. Here, $P_i$ denotes the frequency of a rating in the set of 0, 1 and 2. A higher GPT score indicates better performance.

| Dimensions | Completeness | | | | Precision | | | | Relevance | | | | Sum |
|---|---|---|---|---|---|---|---|---|---|---|---|---|---|
| Model | $P_0$ | $P_1$ | $P_2$ | score | $P_0$ | $P_1$ | $P_2$ | score | $P_0$ | $P_1$ | $P_2$ | score | |
| Bunny-3B (He et al., 2024) | 24.40% | 71.64% | 3.95% | 0.79 | 9.86% | 50.53% | 39.60% | 1.29 | 0.97% | 21.73% | 77.28% | 1.76 | 3.85 |
| LLaVA-v1.5-13B (Liu et al., 2023) | 18.77% | 73.44% | 7.79% | 0.89 | 34.66% | 38.72% | 26.62% | 0.92 | 1.02% | 34.59% | 64.39% | 1.63 | 3.44 |
| mPLUG-Owl2 (Ye et al., 2023) | 19.43% | 65.54% | 14.45% | 1.18 | 30.94% | 43.71% | 24.63% | 1.02 | 3.79% | 26.94% | 68.28% | 1.79 | 3.99 |
| Emu2-Chat (Sun et al., 2023) | 41.25% | 54.33% | 4.42% | 0.63 | 38.11% | 36.41% | 25.48% | 0.87 | 4.12% | 38.61% | 57.27% | 1.53 | 3.03 |
| Qwen-VL-Max (Bai et al., 2023) | 11.64% | 54.08% | 34.08% | 1.22 | 24.26% | 39.14% | 36.22% | 1.11 | 2.53% | 10.97% | 85.64% | 1.82 | 4.15 |
| Gemini-Pro (Google, 2023) | 18.22% | 44.48% | 36.84% | 1.18 | 34.13% | 37.95% | 27.02% | 1.18 | 0.67% | 5.91% | 92.22% | 2.16 | 4.52 |
| GPT-4V (OpenAI, 2023) | 4.09% | 31.82% | 64.09% | 1.60 | 10.40% | 45.12% | 44.44% | 1.34 | 0.18% | 1.69% | 96.35% | 1.94 | 4.89 |
| Co-Instruct (Wu et al., 2024a) | 4.04% | 31.55% | 63.55% | 1.58 | 13.68% | 43.68% | 41.37% | 1.26 | 0.0% | 0.44% | 98.22% | 1.96 | 4.82 |
| Q-Adapt$^{co}$ | 8.97% | 44.22% | 46.79% | 1.38 | 3.82% | 27.15% | 69.02% | 1.65 | 0.0% | 4.17% | 95.8% | 1.96 | 4.98 |

AGIQA-3k (Li et al., 2023a), CGIQA-6k (Zhang et al., 2023d), and KADID-10k (Lin et al., 2019) datasets, which are barely existed in the training process. It underscores the strong generalization ability of Q-Adapt, which can be attributed to the parameter-efficient training approach.

Table 3: The comparison results of quality assessment (SROCC/PLCC).

| Model | KonIQ-10k | SPAQ | LIVE-FB | LIVE-itw | AGIQA-3k | CGIQA-6k | KADID-10k | Average |
|---|---|---|---|---|---|---|---|---|
| LIQE (Zhang et al., 2023c) | 0.897/0.914 | 0.925/0.922 | 0.469/0.541 | 0.868/0.884 | 0.744/0.807 | 0.161/0.197 | 0.675/0.663 | 0.677/0.704 |
| LoDa (Xu et al., 2024) | 0.804/0.844 | 0.892/0.899 | 0.460/0.524 | 0.784/0.820 | 0.687/0.744 | 0.303/0.322 | 0.636/0.649 | 0.653/0.686 |
| LLaVA-v1.5 (Liu et al., 2023) | 0.448/0.460 | 0.563/0.584 | 0.310/0.339 | 0.445/0.481 | 0.285/0.297 | 0.664/0.754 | 0.390/0.400 | 0.444/0.474 |
| mPLUG-Owl2 (Ye et al., 2023) | 0.196/0.252 | 0.589/0.614 | 0.217/0.286 | 0.293/0.342 | 0.473/0.492 | -0.024/-0.032 | 0.541/0.546 | 0.326/0.357 |
| Emu2-Chat (Sun et al., 2023) | 0.664/0.714 | 0.712/0.698 | 0.355/0.341 | 0.597/0.611 | 0.759/0.751 | 0.224/0.269 | 0.841/0.790 | 0.593/0.596 |
| InternLM-XComposer-VL (Zhang et al., 2023a) | 0.564/0.615 | 0.730/0.750 | 0.360/0.416 | 0.612/0.676 | 0.732/0.775 | 0.243/0.265 | 0.546/0.572 | 0.541/0.581 |
| Co-Instruct (Wu et al., 2024a) | 0.839/0.898 | 0.869/0.900 | **0.467/0.584** | 0.839/0.851 | 0.680/0.708 | 0.421/0.438 | 0.762/0.756 | 0.696/0.733 |
| Q-Adapt$^{Co}$ | **0.869/0.898** | **0.916/0.915** | 0.460/0.539 | **0.869/0.897** | **0.739/0.783** | 0.429/0.435 | 0.720/0.711 | **0.714/0.739** |
| Q-Instruct (Wu et al., 2023b) | **0.911/0.921** | 0.901/0.898 | **0.442/0.535** | 0.842/0.840 | 0.700/0.763 | 0.572/0.578 | 0.682/0.683 | 0.721/0.745 |
| Q-Adapt$^{Q}$ | 0.878/0.907 | **0.913/0.916** | 0.440/0.517 | 0.837/0.845 | **0.757/0.789** | **0.593/0.595** | **0.769/0.754** | **0.741/0.760** |

**Parameters and Flops.** Q-Adapt presents an effective tuning strategy that utilizes minimal parameter increases to achieve substantial performance improvements over the baseline model, Bunny-3B, as well as the more parameter-intensive Q-Instruct-8B, thereby offering a more efficient solution for EIQA task adaptation from the well-built LMM.

Table 4: Parameters and FLOPs comparisons for different models, with performance metrics computed on the Q-bench-A1-dev.

| | Q-Instruct-8B | Bunny-3B (LoRA) | Q-Adapt-3B |
|---|---|---|---|
| Flops | 1700G | 656.18 G | 695.32 G |
| Param | 8.2B | 2.78B | 2.98B |
| Performance | 70.23 | 69.57 | **77.19** |

Table 5: Ablation study for instruction-guided visual prompt.

| | Q-bench-A1 (dev) | Q-bench-A1 (test) | Average | Q-bench2-A1 (dev) | Q-bench-A2 (test) | Average |
|---|---|---|---|---|---|---|
| w.o. prompt$^{Q}$ | 74.45 | 75.25 | 74.85 | 52.50 | 51.85 | 52.17 |
| Q-Adapt$^{Q}$ | **77.19** | **77.06** | **77.12** | **55.40** | **55.96** | **55.68** |
| w.o. prompt$^{Co}$ | 75.93 | **75.71** | 75.82 | 76.80 | 76.77 | 76.78 |
| Q-Adapt$^{Co}$ | **76.05** | 76.12 | **76.08** | **77.20** | **78.38** | **77.79** |

## 4.3 ABLATION STUDY

There are three directions to explore the variants of instruction-adaptive visual prompts:

(I) **The Existence of Instruction-guided Visual Prompt.** The effectiveness of instruction-guided visual prompt for Q-Adapt in the Stage 2 training phase is explored in Table 5. In the Table, "w.o. prompt" indicates that only the multimodal connector is trainable. From the results, it is evident that with the assistance of the instruction-guided visual prompt, Q-Adapt achieves a performance gain over training only the connector. It highlights the effect of the instruction-guided visual prompt in adaptively excavating perceptual knowledge required by task instructions.

As shown in Fig. 5, we demonstrate the effectiveness of our proposed instruction-adaptive visual prompting. The visualization results indicate that, for the question answering task, the instruction-adaptive features concentrate on areas specified by the instruction or corresponding to potential answers. In contrast, the visual prompt for the overall quality explanation task typically highlights a broader range of visual details. This demonstrates a dynamic modulation for two EIQA tasks.

Table 6: Comparison with different text encoders for generating instruction-guided visual prompt.

| | Q-bench-A1 (dev) | Q-bench-A1 (test) | Average | Q-bench2-A1 (dev) | Q-bench2-A1 (test) | Average |
|---|---|---|---|---|---|---|
| BERT$^Q$ | 75.72 | 75.65 | 75.69 | 55.10 | 53.15 | 54.12 |
| Q-Former$^Q$ | **77.19** | **77.06** | **77.12** | **55.80** | **55.45** | **55.63** |
| BERT$^{Co}$ | 76.02 | **76.05** | **76.12** | 76.08 | 76.57 | 76.83 |
| Q-Former$^{Co}$ | **76.05** | 76.12 | 76.08 | **77.20** | **78.38** | **77.79** |

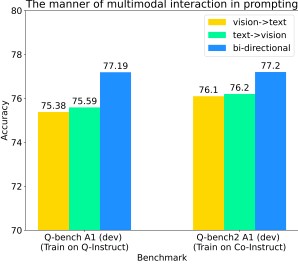

Figure 6: The effect of variants for multimodal interaction.

(II) **The Encoder Structure for V-T Generator.** The analysis for the encoder strure of V-T Generator is shown in Table 6. Utilizing the Q-Former (Dai et al., 2024) can achieve an average accuracy increase of 1.43% on Q-bench-A1 for instruction tuning on Q-Instruct, compared to the BERT (Devlin et al., 2018) structure. It demonstrates that the Q-Former, by introducing learnable queries, can capture high-level semantic information from the instructions, facilitating the extraction of crucial task information.

(III) **Multimodal Interaction.** The multimodal interaction for constructing instruction-adaptive visual prompts is detailed in Fig. 6. It can be observed that the bi-directional interaction between text and visual modalities achieves the highest performance. Additionally, the performance gain from vision-text interaction (*i.e.*, V-T Generator) is lower than that from text-vision interaction (*i.e.*, T-V Prompter), which indicates the importance of mapping textual features into the visual feature space for modulating the original visual features.

(IV) **The Difference with VTC.** VTC Wang et al. (2025) concatenates the additional visual tokens to complete the original visual tokens. We conduct this insert manner like VTC to compare with our spatial-wise modulation in Table 7. The results indicate that concatenating complementary visual tokens is unnecessary when using the uncompressed original visual tokens of Bunny, as the original tokens already provide sufficient information for effective processing.

Table 7: Comparison of Q-bench1-dev performance between our method and VTC.

| Prompting | Q-bench1-dev |
|---|---|
| Ours | 77.19 |
| VTC | 76.99 |

More ablation study for instruction-adaptive visual prompting can be found in **Appendix** A.1

Table 8: Ablation study on progressive instruction tuning on Q-Instruct dataset.

| Training Stages | Tasks | | Module | | | | Q-bench | | |
|---|---|---|---|---|---|---|---|---|---|
| | Quality | Perception | Vision LoRA | LLM LoRA | Connector | Prompt Module | dev | test | Average |
| Stage 1 | ✓ | ✗ | ✓ | ✓ | ✓ | ✗ | **73.51** | **73.31** | **73.41** |
| | ✗ | ✓ | ✓ | ✓ | ✓ | ✗ | 67.96 | 69.83 | 68.89 |
| | ✓ | ✓ | ✓ | ✓ | ✓ | ✗ | 69.57 | 69.89 | 69.73 |
| | ✓ | ✓ | ✓ | ✓ | ✓ | ✓ | 71.30 | 74.38 | 72.84 |
| Stage 2 | ✓ | ✓ | ✗ | ✗ | ✓ | ✓ | **77.19** | **77.06** | **77.12** |
| | ✓ | ✗ | ✗ | ✗ | ✓ | ✓ | 70.10 | 69.40 | 69.75 |
| | ✗ | ✓ | ✗ | ✗ | ✓ | ✓ | 75.59 | 75.45 | 75.52 |
| | ✓ | ✓ | ✗ | ✗ | ✗ | ✓ | 74.85 | 74.11 | 74.48 |
| | ✓ | ✓ | ✗ | ✓ | ✓ | ✓ | 74.45 | 75.98 | 75.21 |

(V) **The Effectiveness of Progressive Instruction Tuning.** We analyze the effect of progressive instruction tuning for training on Q-Instruct in Table 8. Additionally, we examine the impact of task selection for overall quality explanation tasks, as shown in Fig. 7. And we also conduct a comprehensive comparison across different models in Table 9 for joint tuning on two EIQA tasks, two-stage tuning, and our proposed progressive-instruction tuning. More ablation study for progressive instruction tuning can be found in **Appendix** A.2.

**The Task for Instruction Tuning.** For the first stage of instruction tuning (*i.e.*, universal perception knowledge learning stage), the results (the $1^{st}$, $2^{nd}$, and $3^{rd}$ rows of Table 8) show that the performance of joint tuning on both tasks and only tuning on the perception answering task are

lower than tuning on the overall quality explanation task. Also, from Fig. 7, we can see that the performance can be boosted when training on the Quality subset (*i.e.*, overall quality explanation). It reflects the inherent conflicts between the two tasks. For the second stage of instruction tuning (*i.e.*, the instruction-adaptive visual prompting stage), the results ($5^{th}$ and $6^{th}$ rows of Table 8) demonstrate that joint tuning for both tasks yields an average accuracy gain of 1.6% compared to tuning exclusively on the perception answering task. The similar phenomenon is observed in the quality explanation task in Fig. 7, removing the explanation subset results in a performance decline (from 4.98 to 4.95). Additionally, training solely on the quality explanation task in Stage 2 leads to a significant performance decline. This is due to the excessive focus on universal global reasoning, which compromises the model's ability to effectively address question answering tasks. It underscores the significance of achieving synergy between the two EIQA tasks in the second stage.

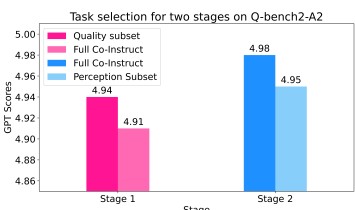

Figure 7: The effect of task selection in progressive instruction tuning for explanation task.

**The Trainable Modules.** For the first stage tuning (*i.e.*, the universal perception knowledge learning stage), the results for trainable modules are shown in the $3^{rd}$ and $4^{th}$ rows of Table 8. The findings reveal that joint tuning on the prompt module results in an average accuracy improvement of 3.11%, demonstrating the effectiveness of instruction-adaptive visual prompts for adapting to different instructions. However, it is still lower than only training on quality explanation tasks, due to the importance of required universal perception knowledge. The results in the second stage (*i.e.*, the instruction-adaptive visual prompting stage) is examined in the $5^{th}$, $7^{th}$, and $8^{th}$ rows of Table 8. We draw two conclusions from the results: Firstly, a trainable multimodal connector is essential for the second stage of instruction tuning, since it plays a critical role in modality alignment. Secondly, a trainable LoRA for the language decoder is unnecessary in the second stage, as the language decoder should remain fixed to preserve the universal perceptual knowledge acquired in the first stage.

Table 9: The comparisons between different models and tuning strategies on Q-bench-A1 (dev), where all methods utilize LoRA for efficient training.

|  | Joint Tuning | Two-Stage Tuning | Progressive Instruction Tuning |
|---|---|---|---|
| LLama-VID-8B (Li et al., 2023b) | 65.55 | 63.81 | 67.49 |
| mPLUG-Owl2-8B (Ye et al., 2023) | 66.69 | 67.76 | 69.03 |
| Bunny-3B (He et al., 2024) | 69.57 | 68.28 | **77.19** |

(III) **Progressive Instruction Tuning across different backbones.**

We present a comprehensive comparison of LLama-VID (Li et al., 2023b), mPLUG-Owl2 (Ye et al., 2023), and Bunny (He et al., 2024) across joint tuning, two-stage tuning, and our proposed progressive instruction tuning on Q-Instruct dataset, as detailed in Table 9. All training strategies utilize LoRA for efficient training. The two-stage tuning approach consists of two phases: initially training on the overall quality explanation task with a trainable multimodal connector for alignment, followed by training on the two EIQA tasks using both the connector and the LLM. Experimental results in the table indicate that progressive instruction tuning yields the best performance, as it effectively mitigates task conflict. In contrast, the two-stage tuning process, which resembles the training strategy of existing LMMs, is inadequate for adapting LMMs to downstream tasks, such as EIQA.

## 5 CONCLUSION

In summary, to alleviate the inherent conflicts in two EIQA tasks (*i.e.*, overall quality explanation, and attribute-wise perception answering), we propose Q-Adapt to adapt LMM as a visual quality perceiver, which is conducted through a perception-oriented instruction tuning strategy, namely, progressive instruction tuning. The progressive instruction tuning consists of the universal perception learning stage for building a powerful base for two tasks, and the instruction-adaptive prompting stage for dynamically adapting visual features for different instructions. By doing this, our Q-Adapt can achieve the synergy between these two EIQA tasks when adapting LMM. Extension experiments on two related benchmarks can illustrate the effectiveness of our Q-Adapt on both overall quality explanation task and attribute-wise perception answering task.

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

# A    APPENDIX

**Training Details.**    The detailed of hyperparameters and modules are listed below: Visual Encoder: siglip-so400m-patch14-384, LLM: phi-2, image resolution: 384, batchsize: 64, learning rate: 2e-5, learning rate schedule: cosine decay, weight decay: 0, warmup ratio: 0.03, gradient accumulation steps: 4, numerical precision: float16, epochs for stage 1: 1, epochs for stage 2: 1, optimizer: AdamW, deepspeed stage: 2.

Following the pioneering works of LMM paradigm (Wu et al., 2023b) of finetuning strategy and model architecture, we inherit weights from the Bunny-3B of instruction version to apply continual instruction tuning to downstream EIQA tasks.

In the progressive instruction tuning approach applied to the Q-Instruct dataset, the first stage solely focuses on the overall quality explanation task to acquire universal knowledge. The second stage involves joint tuning across the full Q-Instruct dataset. For the Co-Instruct dataset, given that the baseline model, Bunny-3B, has not been exposed to multiple images for vision question answering, we transform the attribute-wise perception answering task data into chain-of-thought quality data (i.e., multi-turn conversations). This data is then combined with the overall quality explanation task data to fulfill the requirements for universal knowledge acquisition. In the second stage, we train our Q-Adapt model on the entire Co-Instruct dataset. In all stages, the first stage focuses solely on training the LoRA of the visual encoder, the language decoder, and all multimodal connector. The second stage is dedicated exclusively to training the prompt module and the multimodal connector.

**Evaluation Metric.**    For the attribute-wise perception answering task, we apply accuracy as the metric to measure the performance. For overall quality explanation task, we adopt 5-round GPT evaluation score for comparison between our generated explanation and ground-truth explanation on completeness, precision, and relevance. For quality assessment task, We adopt two widely used criteria for performance evaluation: Pearson linear correlation coefficient (PLCC) and Spearman rank order correlation coefficient (SROCC). A higher value for these coefficients indicates a stronger correlation with quality annotations.

## A.1 MORE ABLATION STUDY FOR INSTRUCTION-ADAPTIVE VISUAL PROMPTING

There are three directions to explore the variants of instruction-adaptive visual prompt for the overall quality explanation task.

**The Existence of Instruction-adaptive Visual Prompt.**    As demonstrated in Fig. 8, our instruction-adaptive visual prompt (*i.e.*, the $3^{rd}$ bar) enables the Q-Adapt to outperform the baseline without visual prompt (*i.e.*, the $1^{st}$ bar) on Q-bench2-A2, achieving a gain of 0.23 on GPT Score. With the same results in Table 5 for attribute-wise perception answering task, it underscores the effectiveness of instruction-adaptive visual prompts in alleviating conflicts in Explainable Image Quality Assessment (EIQA) tasks and in promoting the synergy between the two tasks.

**The Encoder Structure for V-T Generator.**    As depicted in Fig. 8, the Q-former (*i.e.*, the $3^{rd}$ bar) for building V-T Generator in instruction-adaptive visual prompt can surpass the Bert structure (*i.e.*, the $2^{nd}$ bar). With the same results in Table 6 for the attribute-wise perception answering task, it is evident that the Q-former structure is more suitable for instruction understanding in the EIQA tasks.

**Multimodal Interaction.**    The results are represented in Fig. 9. For the overall quality explanation task, it is observed that both bi-directional interaction and text-vision interaction outperform vision-text interaction with the same GPT score evaluation. As presented in Fig. 6, the bi-directional interaction achieves higher accuracy compared to both text-vision and vision-text interactions on attribute-wise perception answering task. This underscores the universality of bi-directional interaction in facilitating both types of EIQA tasks.

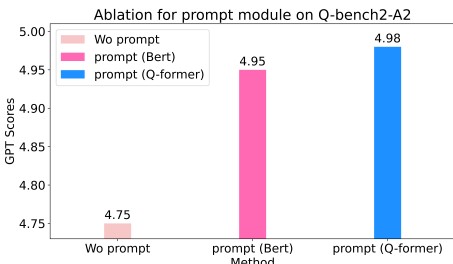
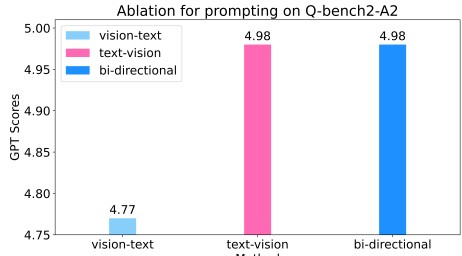

Figure 8: The effect of instruction-adaptive visual prompt with different structure for training on Co-Instruct.

Figure 9: The effect of the manner for mutlimodal interaction for training on Co-Instruct.

Table 10: Ablation study on progressive instruction tuning on Co-Instruct dataset.

| Training Stages | Tasks | | Module | | | | Q-bench2 | | |
| | Quality | Perception | Vision LoRA | LLM LoRA | Connector | Prompt Module | A1 dev | A1 test | A2 |
|---|---|---|---|---|---|---|---|---|---|
| | ✓ | ✗ | ✓ | ✓ | ✓ | ✗ | **75.90** | **75.58** | **4.94** |
| Stage 1 | ✗ | ✓ | ✓ | ✓ | ✓ | ✗ | 70.60 | 73.77 | 3.18 |
| | ✓ | ✓ | ✓ | ✓ | ✓ | ✗ | 73.50 | 73.27 | 4.91 |
| | ✓ | ✓ | ✓ | ✓ | ✓ | ✓ | 71.60 | 72.17 | 4.99 |
| | ✓ | ✓ | ✗ | ✗ | ✓ | ✓ | **77.20** | **78.38** | **4.98** |
| Stage 2 | ✗ | ✓ | ✗ | ✗ | ✓ | ✓ | 75.79 | 75.45 | 4.95 |
| | ✓ | ✓ | ✗ | ✗ | ✗ | ✓ | 77.30 | 77.47 | 4.83 |
| | ✓ | ✓ | ✗ | ✓ | ✓ | ✓ | 76.10 | 76.37 | 4.92 |

## A.2 MORE ABLATION STUDY FOR PROGRESSIVE INSTRUCTION TUNING FOR TRAINING ON CO-INSTRUCT.

We also analyze the effect of progressive instruction tuning in Table 10 for Q-Adapt training on Co-Instruct.

**Task Selection for Progressive Instruction Tuning.**    For the first stage of instruction tuning (*i.e.*, universal perception knowledge requiring stage), it is observed that training on the full Co-Instruct

dataset results in the lower performance (*i.e.*, 4.91 on the $3^{rd}$ row) than training on the quality subset (*i.e.*, 4.94 on the $1^{st}$ row) for the overall quality explanation task. It suggests that task conflicts lead to a bias towards attribute-wise specific knowledge at the expense of comprehensive reasoning capabilities. The loss of this comprehensive reasoning ability also contributes to reduced performance from 75.90 (*i.e.*, the $1^{st}$ row) to 73.50 (*i.e.*, the $3^{rd}$ row) on attribute-wise perception answering task, through the comparison between training on quality data and training on full Co–Instruct. For the second stage (*i.e.*, instruction-adaptive visual prompting stage), we can see that removing the quality data (*i.e.*, the overall quality explanation task) will result in a performance decline from 4.98 (*i.e.*, the $5^{th}$ row) to 4.95 (*i.e.*, the $6^{th}$ row) on the overall quality explanation task. It reveals the importance of synergy between two EIQA tasks.

**Trainable Modules Selection for Progressive Instruction Tuning.** For the first stage of instruction tuning, we can see that combination with the prompt module obtains the highest performance (*i.e.*, 4.99, the $4^{th}$ row), which is attributed to the enhanced task adaptation ability by instruction-adaptive visual prompting. However, the performance of the combination with the prompt module in the first stage is a little low on perception answering task (*i.e.*, 71.6, the $4^{th}$ row). Therefore, it reveals that the combination with the prompt module in the first stage is not optimal for universal knowledge acquisition. In the second stage of instruction tuning, our observations indicate that both incorporating LoRA and removing the multimodal connector result in a performance decline (*i.e.*, from 77.20/78.38/4.98 to 77.30/77.47/4.83, and from 77.20/78.38/4.98 to 76.10/76.37/4.92 ) on both overall quality explanation task and attribute-wise perception answering task.

### A.3    Samples for Two EIQA Tasks.

The comprison between existed LMMs and our proposed Q-Adapt on two EIQA tasks (*i.e.*, the overall quality explanation task, and the attribute-wise perception answering task). Our proposed Q-Adapt can generate more accurate response, benefiting from the reduction of task conflicts and the enhanced synergy between the two tasks, achieved through progressive instruction tuning. Due to the task conflict alleviation by our proposed progressive instruction tuning and instruction-aware visual prompt tuning, our Q-Adapt can ensure the accuracy of fine-grained low-level perception.

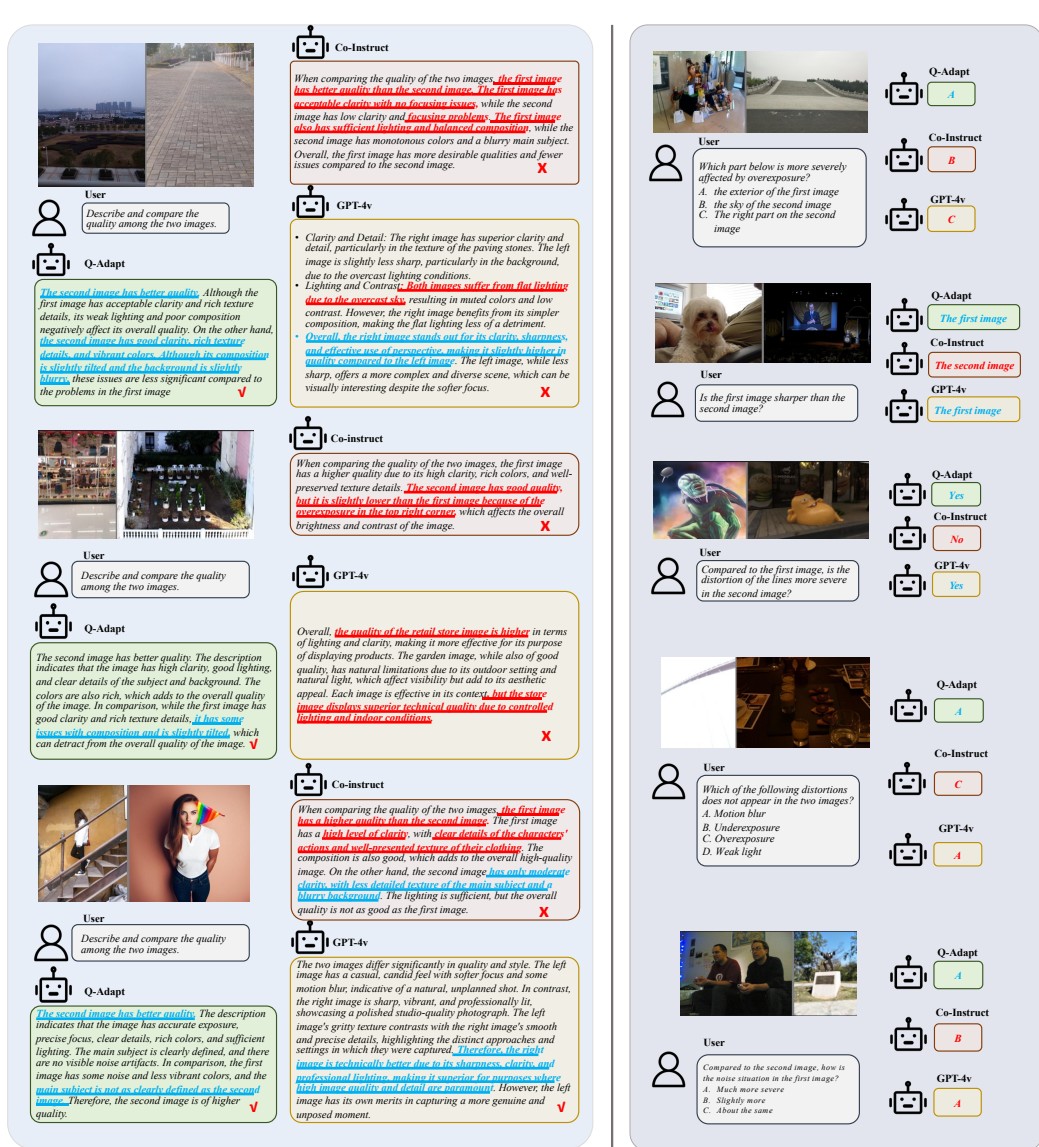

Figure 10: The comparison of existing LMM and our Q-Adapt on two EIQA tasks (*i.e.*, the overall quality explanation task and the attribute-wise perception answering task).

