# OpenReview forum: "Q-Adapt: Adapting LMM for  Visual Quality Perceiver with Progressive Instruction Tuning"
_ICLR.cc/2025/Conference — Submitted to ICLR 2025_

### Official Review · Reviewer_YtoC · 2024-10-22

**Soundness:** 3
**Presentation:** 3
**Contribution:** 2
**Rating:** 5
**Confidence:** 3

**Summary:**

This article proposes progressive instruction tuning to address the Explainable Image Quality Assessment problem. This method can solve the conflict problem when two types of datasets are jointly trained and achieve better multi-task training. The article also proposes bidirectional interaction between instruction and visual feature to further improve model performance. The proposed method has achieved excellent performance on multiple EIQA tasks.

**Strengths:**

1. This article is well written.
2. The proposed method performs well on some datasets.

**Weaknesses:**

1. In Table 1, Q-Adapt-3B is worse than Co-Instruct-8B with Co-Instruct dataset for training. Can you explain this phenomenon? Considering that Co-Instruct have more data than Q-Instruct. Does this mean that your method will fail (i.e., worse than Co-Instruct-8B) when the amount of data is large enough? If it is due to the amount of model parameters, is it possible to conduct a comparative experiment using the same LLM as Co-Instruct-8B?
2. Lack of ablations. From Table 7, i notice that joint training (69.73) is better that Perception training (68.89) but worse than Quality training (73.41) in Stage 1. However, there is a lack of only quality training experiments in Stage 2. So, i wonder if joint training is the best method in stage 2? What about joint training v.s. quality training in stage 2?

**Questions:**

Please comment the two points in the weakness section.

---

### Official Review · Reviewer_ntj8 · 2024-10-29

**Soundness:** 3
**Presentation:** 2
**Contribution:** 2
**Rating:** 6
**Confidence:** 4

**Summary:**

The paper introduces Q-Adapt, a framework designed to adapt Large Multi-modal Models (LMMs) for Explainable Image Quality Assessment (EIQA). It tackles the challenge of handling two conflicting EIQA tasks: overall quality explanation and attribute-wise perception answering. To resolve these conflicts, the authors propose a progressive instruction tuning approach that consists of two stages. In the first stage, universal perception knowledge is acquired using a parameter-efficient method like LoRA to fine-tune the model for both tasks. In the second stage, instruction-adaptive visual prompting is introduced to allow the model to dynamically adapt to task-specific instructions, improving the synergy between vision and language features. The experimental results show that Q-Adapt significantly improves performance across benchmarks, efficiently balancing both EIQA tasks and leading to better visual perception and reasoning.

**Strengths:**

[1] The paper provides extensive experiments, which convincingly demonstrate the effectiveness of the proposed method.

[2] The approach achieves performance on par with 8B models while only using a 3B model, which is highly encouraging.

[3] The method is simple and straightforward, making it easy to implement and understand.

[4] The method offers valuable insight into the task conflicts within EIQA, shedding light on how to address these challenges effectively.

**Weaknesses:**

[1]The motivation section needs improvement. In Line 51, the conflict between the two EIQA tasks is mentioned without explaining why this conflict arises or giving examples. This should be clarified in the introduction rather than in Line 220.

[2]It’s unclear what the proposed method gains from the perception answering task in the second phase. The paper stresses the importance of the explanation task and highlights the negative impact of the perception answering task during joint training. If this is the case, why include the perception answering task at all? The authors should provide a detailed analysis of the knowledge required for both tasks, as well as how they benefit from and conflict with each other.

[3]The proposed V-T Generator and T-V Prompter are central to the paper, but their approach seems similar to [1]. While this might be new for the IQA field, the contribution to the broader CV and MLLM communities appears limited.

[4]The results in Figure 3 are confusing. The caption refers to the quality explanation task, but Line 216 and the y-axis suggest these are results for the attribute-wise perception answering task. This needs clarification.

[5]In the experiments presented in Figure 3, was the data volume kept consistent across the two tuning settings? Assuming that Figure 3 indeed represents results for the attribute-wise perception answering task, why does the model tuned specifically for this task perform the worst? Could this be attributed to differences in data volume?

[6]The reviewer suggests merging the paragraph starting at Line 244 in Section 3.3 with Section 3.4. The current organization may mislead readers into thinking that the connector is responsible for adaptively selecting the required perceptual knowledge based on task instructions

[7]How significant is the impact of using MLP versus Q-Former as the connector on the final results?

[8]Table 7 lacks an ablation study that focuses solely on fine-tuning with the quality explanation task in the second stage. This would help evaluate whether including the attribute-wise perception answering task is beneficial to model training.

[9]Based on the results reported in Table 2, the sum score for Qwen-VL-Max should be 5.18, not the current value. The reviewer recommends carefully verifying the reported data throughout the paper.

[10]Including more comparisons with other MLLM performance benchmarks would further strengthen the paper. For instance, comparisons with models such as LLaVA-OneVision [2], LLaVA-NeXT-Interleave-7B [3], and Qwen2-vl [4] would be highly beneficial.

[11]Minor Weaknesses:
- The use of "(i)" and "(ii)" in the second paragraph of the Introduction appears multiple times, which disrupts the flow and readability. The reviewer suggests varying the symbols when listing multiple points.
- There is a typo in Table 6, second row: "$Q$-$Former^{Co}$" should be "$Q$-$Former^Q$."

[1] Instruction Tuning-free Visual Token Complement for Multimodal LLMs.

[2] Llava-OneVision: Easy Visual Task Transfer.

[3] Llava-Next-Interleave: Tackling Multi-Image, Video, and 3D in Large Multimodal Models.

[4] Qwen2-VL: Enhancing Vision-Language Model’s Perception of the World at Any Resolution.

**Questions:**

[1] What are the computational costs of using the V-T generator and T-V prompter modules, respectively?

[2] Are there some examples of failed feature visualizations?

[3] The overall quality of IQA images is related to both global and local information. The Visual Prompt (Task 2) in Figure 4 seems to rely too much on global information, which seems unusual. Could you explain why this might be the case?

---

> ### Comment · Reviewer_ntj8 · 2024-11-27
>
> Thank you for the reply, authors.
>
> > [3]The proposed V-T Generator and T-V Prompter are central to the paper, but their approach seems similar to [1].
>
> I still find the explanation here unconvincing. The visual information in VTC is text-relevant rather than "text-irrelevant," which reinforces my view that the proposed components share similarities with the VTC method [1]. Specifically, the V-T Generator appears comparable to the Q-Former in VTC, while the T-V Prompter resembles the VTC module in VTC.
>
> > [10] Including more comparisons with other MLLM performance benchmarks would further strengthen the paper.
>
> Adding these results and incorporating them into Table 1 would improve the completeness of the evaluation.
>
> Overall, the authors have addressed most of my concerns, and the method achieves competitive performance. As a result, I will raise my score. However, my primary concern remains the core technical contribution of the paper, as the designs of the V-T Generator and T-V Prompter lack sufficient novelty.

---

### Official Review · Reviewer_fuTc · 2024-10-30

**Soundness:** 2
**Presentation:** 2
**Contribution:** 2
**Rating:** 5
**Confidence:** 5

**Summary:**

This work proposes to adapt MLLM for visual quality assessment (as textural output) with a two-stage instruction tuning.

**Strengths:**

1. The use of MLLM for visual quality assessment is a rising research direction that is worth deep investigation.
2. The related work section, for example, the categorization of MLLM-based visual quality assessment, is very well written.
3. The experimental results look good.

**Weaknesses:**

1. The term "progressive" is a little misleading; it is simply a two-stage training method.

2. The motivation for this work is unclear. The authors claim that the two examined tasks are conflicting, which is not well justified. From the reviewer's perspective, the two tasks could complement each other: a holistic understanding of visual quality aspects of the image can enhance more detailed quality assessment tasks involving local image analysis, and vice versa. This indeed motivates the authors to propose the two-stage training method.

3. The use of LoRA in the first stage of training is not clearly explained. Why is parameter-efficient fine-tuning (such as LoRA) necessary? Which subset of parameters is subject to LoRA fine-tuning, and how are these parameters identified?

4. The information flow in stage two (Figure 2) appears somewhat redundant. For example, both raw visual features and processed visual features (by the V-T generator) are sent to the T-V prompter, and then combined with the raw features to feed into the language model. Is such a complex design necessary? Further justification would be appreciated.

5. The generated quality-relevant textual descriptions should be evaluated for both correctness and diversity. Template-like textual outputs are unlikely to be perceived as explainable.

6. The experimental setups need clearer descriptions. For instance, how are the competing methods implemented?

7. How were the visualizations in Figure 4 generated, and how should these results be interpreted and compared?

**Questions:**

The authors should work on Points 2, 3, 4, and 5 to raise the reviewer's rating.

---

### Official Review · Reviewer_Rdod · 2024-11-01

**Soundness:** 1
**Presentation:** 2
**Contribution:** 2
**Rating:** 5
**Confidence:** 4

**Summary:**

This work proposed a paradigm for perception-oriented instruction tuning named Q-Adapt, aiming to alleviate the conflict between overall quality explanation task and attribute-wise perception answering task. The authors designed a two-stage training strategy. In stage one, a LMM is finetuned to obtain a powerful base for two tasks; in stage two, some parameters in the LMM are frozen while the other parameters are finetuned to drive LMM more focus on the task instruction. In the second stage, a V-T generator and a T-V prompter are designed to achieve the bi-directional multimodal interactions.

**Strengths:**

In terms of originality, this paper proposes a new method; In terms of quality, based on the experimental results provided by authors, the proposed method can only improve the performance in a small number of experiments, thus its superiority is worth reconsidering. In terms of clarity, the authors describe the implementation details of the proposed method in detail, but the description of the experimental settings is not clear enough, which is easy to mislead the reader. In terms of importance, if the proposed method could actually improve the performance of LMM on two tasks, it would be a promising method which worthes further researching.

**Weaknesses:**

1.	The symbols in some equations are not explained clearly, such as n and m near line 196 representing the dimension of feature space.
2.	The legend is completely consistent with the abscissa label in the Figure 3, so the authors are advised to remove the legend.
3.	It is suggested that the authors add an experiment in Sec 3.2, using the same method as in Figure 3 to perform finetune on another task, attribute-wise perception answering task, to prove that joint finetune will affect the performance which is mentioned in the conclusion 1 given by the authors in Sec 3.2. At present, it can only be proved that adding perception negatively affects the overall quality task.
4.	The left values are exactly the same for Equ.3 and Equ.7. Although the author declares "update" above Equ.7, it is also recommended to use different symbols for the left values of Equ.3 and Equ.7 to avoid confusion.
5.	It seems that the authors forgot to mark the better results as bold in the top right grid of Table 6.
6.	The sentence starting with “We analyze the effect of progressive...” at line 468 does not seem to belong to the previous section titled “Multimodal Interaction”; it appears that a title is missing here.
7.	Authors are suggested to add experiments similar to Figure 3 in the experiment section, using Quality, Perception, Joint and proposed methods to finetune the model separately, and test it on two EIQA tasks, in order to verify that the proposed method can "improve synergy between the two EIQA tasks and enhances the multifaceted explanations of IQA".

**Questions:**

1.	Why does the author only analyze the experimental results of Q-bench-A1 in Table 1 and not those of Q-bench2-A1?
2.	What does a score of 0, 1, and 2 mean in Table 2, and how is the GPT score calculated?
3.	In Table 1 and Table 3, the proposed method is even worse than vanilla LMM in many experiments. Can the author explain the reasons for this phenomenon?

---

### Meta-Review · Area_Chair_LE5Q · 2024-12-17

**Metareview:**

The authors present a method to improve the ability of LMMs to explain image quality assessments, e.g., answer prompts like "Describe and compare the quality of these two images" or "Is the first image more blurred than the second image?" To this end, the authors invent a two-stage training strategy.

Even after the rebuttal, the authors do not sufficiently make clear the novelty of their approach. In particular, their bi-directional multimodal interaction and progressive instruction fine-tuning schemes are similar to prior publications, as pointed out by the reviewers.

**Additional Comments On Reviewer Discussion:**

Three of four reviewers criticize the novelty of the work, in particular citing that their bi-directional multimodal interaction and progressive instruction fine-tuning schemes are similar to prior publications. Even after elaboration by the authors, they are not able to convince the reviewers. Reviewer Rdod also finds the methods' "superiority worth reconsidering" given the empirical results. The authors do not respond to this.

---

### Decision · Program_Chairs · 2025-01-22

Reject